# Development of Multiplex PCR and Melt–Curve Analysis for the Molecular Identification of Four Species of the Mullidae Family, Available in the Market

**DOI:** 10.3390/genes14050960

**Published:** 2023-04-23

**Authors:** Ioannis A. Giantsis, Maria Tokamani, George Triantaphyllidis, Stella Tzatzani, Emmanuella Chatzinikolaou, Athanasios Toros, Anastasia Bouchorikou, Evanthia Chatzoglou, Helen Miliou, Joanne Sarantopoulou, Georgios A. Gkafas, Athanasios Exadactylos, Raphael Sandaltzopoulos, Apostolos P. Apostolidis

**Affiliations:** 1Department of Animal Science, Faculty of Agricultural Sciences, University of Western Macedonia, 53100 Florina, Greece; igiants@agro.auth.gr; 2Department of Molecular Biology and Genetics, Democritus University of Thrace, 68100 Alexandroupolis, Greece; tokamanimaria@hotmail.com (M.T.);; 3Laboratory of Fish & Fisheries, Department of Animal Production, School of Agriculture, Faculty of Agriculture, Forestry and Natural Environment, Aristotle University of Thessaloniki, 54124 Thessaloniki, Greece; 4Laboratory of Applied Hydrobiology, School of Animal Biosciences, Agricultural University of Athens, 11855 Athens, Greece; gvtrianta@aua.gr (G.T.); echatzoglou@aua.gr (E.C.); elenmi@aua.gr (H.M.); 5Hydrobiology-Ichthyology Lab, Department of Ichthyology and Aquatic Environment, School of Agricultural Sciences, University of Thessaly, 38446 Volos, Greece; saradopo@uth.gr (J.S.); gkafas@uth.gr (G.A.G.); exadact@uth.gr (A.E.)

**Keywords:** *Mullus barbatus*, *Mullus surmuletus*, *Upeneus moluccensis*, *Pseudopeneus prayensis*, mPCR, mRT-PCR, CO1, CYTB

## Abstract

The authentication of food products and the verification of their identity are of major importance for consumers. Food fraud through mislabeling is an illegal practice consisting of the substitution of an expensive food product by a relatively cheaper one, misleading false labelling of their origin and adulteration in processed or frozen products. This issue is particularly of high importance concerning fish and seafood, which are easily adulterated primarily due to difficult morphological identification. Fish species of the Mullidae family are considered among the most high-valued seafood products traded in Greece and Eastern Mediterranean in general, in terms of the price and demand. Specifically, the red mullet (*Mullus barbatus*) and the striped red mullet (*Mullus surmuletus*) are both indigenous in the Aegean (FAO Division 37.3.1) and the Ionian (FAO Division 37.2.2) Seas, with high levels of consumers’ preferences. However, they could be easily adulterated or misidentified by the invasive Aegean Sea Lessepsian migrator goldband goatfish (*Upeneus moluccensis*) as well as by the imported West African goatfish (*Pseudupeneus prayensis*). Keeping this in mind, we designed two novel, time-saving and easy-to-apply multiplex PCR assays and one multiple Melt–Curve analysis real-time PCR for the identification of these four species. These methodologies are based on species-specific primers targeting single nucleotide polymorphisms (SNPs) detected via sequencing analysis of the mitochondrial cytochrome C oxidase subunit I (CO1) and of the cytochrome b (CYTB) genes in newly collected individuals, with additional comparison with congeneric and conspecific haplotypes obtained from the GenBank database. Both methodologies, targeting CO1 or CYTB, utilize one common and four diagnostic primers, producing amplicons of different length that are easily and reliably separated on agarose gel electrophoresis, yielding a single clear band of diagnostic size for each species or a certain Melt–Curve profile. The applicability of this cost-effective and fast methodology was tested in 328 collected specimens, including 10 cooked samples obtained from restaurants. In the vast majority (327 out of the 328) of the specimens tested, one single band was produced, in agreement with the expected products with a single exception a *M. barbatus* sample that was identified as *M. surmuletus*, the identity of which was confirmed using sequencing, indicating erroneous morphological identification. The developed methodologies are expected to contribute to the detection of commercial fraud in fish authentication.

## 1. Introduction

Modern consumer rights demand the authentication and verification of seafood species labelling as well as their geographic origin (traceability), that together with other factors, ensure food safety [1]. Food products mislabeling may refer to the substitution of species of high value with lower cost alternative ones, which are usually morphologically similar, or cannot be morphologically identified due to processing that may alter their visual characteristics [2]. Particularly, fish and seafood are among the most easily adulterated foods, mainly due to the severe changes in their morphological characteristics during procedures, such as filleting, that require the removal, mainly of morphological features that serve as morphological identification keys, thus elevating them to the top of the adulterated foods list [1]. Even when the morphological characteristics are not altered and the fish are traded as caught, several of them, usually closely related ones, are difficult to distinguish if the consumer does not have sufficient experience in fish taxonomy. It should be emphasized that the adulteration of seafood products and the marketing of fish with false labeling, in addition to deceiving the consumer, involve the risk of the presence of unreported allergens or toxic ingredients, which may cause serious public health issues [3]. Adding or substituting different and occasionally cheaper types of fish for more expensive ones is a well-known form of fraud in the food industry. Nevertheless, this may happen accidentally due to lack of expertise, or when small-size specimens, are traded. Interestingly, this type of fraud is estimated to cause damages of approximately 8 to 12 billion euros per year [4].

Greece is characterized by high seafood consumption rates, ranging from 10 to 20 kg per capita annually [5], encompassing a great number of angled marine fish species, i.e., more than 60 species originating from fisheries production [6]. Greek consumers show high preference for marine fish as an integral part of the Mediterranean diet, yielding great profits to those involved in their marketing and a great nutritional benefit for consumers [7]. The Mullidae family of fishes holds a dominant position in the Greek fish market.

The Mullidae family are exclusively marine organisms, with sizes up to 60 cm although body length is usually much shorter [8]. There is a Minimum Conservation Reference Size (MCRS) of 11 cm according to Annex IX of Regulation (EU) 2019/1241 of the European Parliament and of the Council of 20 June 2019. They contain two long barbels, i.e., chemosensory organs useful in probing the sand or opening reef holes to predate benthic marine invertebrates or smaller fish. Their colors are often bright exhibiting high, not always species-specific variability. Although several have high commercial value as human food in many countries of the world, including Greece, their exact value depends on the different species. Three genera belonging to Mullidae (*Mullus, Upeneus* and *Pseudopeneus*) inhabit the Mediterranean (FAO 37 area), within which, two exotic species have been recorded (*U. moluccensis* and *U. pori*) [9].

The members of Mullidae family constitute morphologically a percomorph syngnathiform-scombriform clade carrying two hyoid barbels in their head that are used for the detection of food [8]. The genus *Mullus* is distributed throughout the Mediterranean, the Black Sea, and parts of the eastern coasts of the Atlantic Ocean [9]. The genus *Upeneus* is a tropical taxon originating from the Red Sea, recently found in the Mediterranean as well, whereas the genus *Pseudopeneus*, which constitutes a larger in length fish than the other genera of the family, inhabits only the Eastern Atlantic [9,10]. All three genera are middle-sized fish, representing high-value traded seafood products.

The red mullet (*M. barbatus*) and the stripped red mullet *(M. surmuletus*) are the two native species of the family in the Aegean and the Ionian Sea, in the Eastern Mediterranean. However, these two species may be adulterated or accidentally mislabeled with fish of the genus *Upeneus* commonly known as goldband goatfish, which constitute a Lessepsian invasive species mostly in the southeast Mediterranean. The landings of this fish species are either mixed and sold together with *M. barbatus* and *M. surmuletus*, or sold separately under common names that mislead consumers into thinking that they are only exceptional ecotypes of the two *Mullus* species. In addition to this adulteration, seafood traders have imported the eastern Atlantic species *P. prayensis*, which is more phylogenetically related with the genus *Mullus* than *Upeneus* sp. [11], to meet consumers’ demand for *M. surmuletus* in national markets of several Mediterranean countries. The economic value of the two latter is considerably lower compared to that of the two indigenous species. Although there are discrete morphometric characters for these species [12], morphological identification is occasionally puzzling, on account of either bad preservation or small size.

In these cases, molecular techniques, such as DNA barcoding, can provide a very effective alternative for reliable species identification. Furthermore, academia worldwide is encouraging researchers to deposit data in reference banks so as to be utilized in future studies. However, as sequencing is a comparatively time-consuming method, especially for laboratories that do not have appropriate sequencing facilities, it is not cost-efficient and not appropriate for large sample numbers. Even for well-equipped laboratories, sequencing methodologies are laborious and expensive and require great technical proficiency [13,14].

Thus, user-friendly molecular techniques that could be implemented in standard not sophisticated equipment would be very helpful towards the fast routine identification of large numbers of food product particles [1].

In this context, the aim of the present study was to develop a fast, reliable, and low-cost method for the identification of the fish species of the family Mullidae traded in the Eastern Mediterranean. For this purpose, two simple multiplex PCR and one real-time PCR Melt–Curve analysis techniques were designed, avoiding enzymatic or sequencing analyses. In an attempt to optimize the developed methodologies targeting all kinds of seafood, an additional scope of the study was to examine the methodology in cooked (olive oil fried) fish.

## 2. Materials and Methods

### 2.1. Sampling, Morphological Identification, and DNA Extraction

Three hundred eighteen (318) specimens were collected from several marine areas, located in the Eastern Mediterranean and Black Seas, during a 2-year period in 2020 and 2021 (Figure 1, Table 1). Sampling locations for *M. surmuletus* and *M. barbatus* (Figure 2) were selected in an effort to cover all representative areas of the native and invasive Mullidae species in the Ionian, Aegean and Black Seas as well as from Cyprus. However, samples collected from Burgas, Bulgaria and Constanta, Romania were only *M. barbatus*, since *M. surmuletus* is not present in the Black Sea, a hypothesis tested also in the present study. *U. moluccensis* (Figure 2), which has been detected in the Eastern Mediterranean as an invasive species [10], was collected from Cyprus. Finally, *P. prayensis* (Figure 2) specimens were purchased from the market, originating from the Atlantic Ocean (FAO 34 area, Rome, Italy). Apart from the aforementioned samples, 10 *M. barbatus* fried samples were also received from restaurants and included in the analysis. Furthermore, gDNA samples from five *Trachurus* sp. were included in the analyses to validate the specificity of species-specific primers. The preservation of samples before further procedures was performed either as whole fish in the freezer (−20 °C) or a small fin tissue was immersed in absolute ethanol. No morphological identification was conducted; instead, the identity proposed by the professional fishermen and by the traders was kept, in order to detect potential cases of commercial fraud. Genomic DNA extraction was obtained from the dorsal fin using the NucleoSpin Tissue kit (Macherey-Nagel, Düren, Germany), following the manufacturer’s instructions. The concentration and purity of the genomic DNA were determined using a Nanodrop 2000 spectrophotometer (Thermo Scientific, Waltham, MA, USA), and the DNA integrity was tested via electrophoresis in 1% *w*/*v* agarose gel.

### 2.2. Sanger Sequencing and Primer Design

To detect and differentiate the selected species, CO1 (cytochrome C oxidase subunit I) and CYTB (cytochrome b) genes regions were used as markers due to the presence of inter-species variations that may serve as diagnostic landmarks allowing discrimination of the four species. PCR was conducted in 40 representative individuals (10 from each species) using the universal primers for CO1 gene [FISHCO1LBC: 5′-TCAACYAATCAYAAAGATATYGGCAC-3′, FISHCO1HBC: 5′-ACTTCYGGGTGRCCRAARAATCA-3′] which produce amplicon of 655bp length [15] and for CYTB gene [L14841: 5′-AAAAAGCTTCCATCCAACATCTCAGCATGATGAAA-3′, H15149: 5′-AAACTGCAGCCCCTCAGAATGATATTTGTCCTCA-3′] which amplify a 464 bp amplicon [16]. The amplicons were sequenced using both forward and reverse universal primers in an ABI Prism 3730XL automatic capillary sequencer in CEMIA Lab (Larissa, Greece). Sequences were processed using MEGA 7 [17] and compared with sequences of closely-related species in BOLD database PDN [18] as well as in the NCBI nucleotide sequence database, whereas those that represented novel haplotypes were submitted in the GenBank database and received accession numbers OK247993-OK248002. Sequences were aligned using the MUSCLE algorithm, embedded in MEGA tool, and specific single-nucleotide polymorphisms were utilized to design allele-species specific primers for each of the four of the studied species: *Mullus barbatus*, *Mullus surmuletus*, *Upeneus moluccensis*, and *Pseudupeneus prayensis* in comparison also with all available CO1 and CYTB sequences in the aforementioned databases. In order to develop a multiplex qPCR assay using a minimum set of primers, one consensus common primer and four species-specific primers were designed. The 3′ end of each species-specific primer was designed to target the diagnostic SNP of each species. Moreover, the target amplicons were generated as different sizes that are species specific. The melting temperatures (Tm) of the expected amplicons of the qPCR-Melt–Curve were adjusted to differ by at least one degree. The final sequences of the selected primers were synthesized using Eurofins Genomics and are demonstrated in Table 2 and Figure 3a,b.

### 2.3. Multiplex PCR Reaction

The multiplex PCR was performed using the FastGene Taq 2 × Ready Mix kit (Nippon Genetics Europe), according to the manufacturer’s instruction. The reaction was optimized and each reaction mixture included: (i) 10 μL of 2 × FastGene Ready Mix, (ii) 0.6 pmol of each primer and water up to the final volume, in a 20 μL reaction. The amplification process was conducted as follows: initial denaturation at 95 °C for 3 min, 38 cycles of denaturation at 95 °C for 15 s, primer annealing at 54 °C 30 s and extension at 72 °C for 30 s, followed by a final extension at 72 °C for 5 min. At the end, PCR products were visualized using electrophoresis on a 2% *w*/*v* agarose gel containing ethidium bromide (EtBr, 0.5 μg/mL) in 0.5× TBE buffer and observed under UV light (ChemiDoc Imaging Systems, Life Science Research, Bio-Rad, Hercules, CA, USA).

### 2.4. Real-Time PCR Reaction and Melt–Curve Analysis

Real-time PCR (qPCR) was performed on a StepOne PCR System in MicroAmp^®^ Fast Optical 48-well reaction plates (Applied Biosystems, Waltham, MA, USA) using KAPA SYBR^®^ FAST Universal Kit [KK4601] (Sigma-Aldrich, St. Louis, MO, USA). The reaction was optimized and each reaction mixture included: (i) 10 μL of 2× KAPA SYBR Fast qPCR Universal buffer, (ii) 0.4 μL High ROX dye, (iii) 1 μL diluted genomic DNA (typically 1–10 ng of DNA), and (iv) 0.2 μM of each primer, in a 20 μL reaction. The amplification process was conducted as follows: initial denaturation at 95 °C for 3 min, 40 cycles of denaturation at 95 °C for 15 s and primer annealing and extension at 60 °C for 30 s. Each experiment included two no-template controls for primer mix. The reproducibility of the method was checked by performing each reaction at least three times. PCR products were evaluated using Melting–Curve Analysis using the following steps: denaturation at 95 °C for 15 s, annealing at 60 °C for 1 min and gradual denaturation from 60 °C to 95 °C by measuring fluorescence emission per 0.3 °C every 15 s.

The specificity of each primer mix was cross-validated via sequencing. The sequences were processed using UGENE version 45.1 [19] and compared with sequences of closely related species in the BOLD database. The positive control qPCR reactions included the targeted template (10 ng). Two no-template reactions for each primer set, served as negative controls. To test for cross-reactivity of the different set of primers, a third type of reaction was designed. This qPCR included the consensus primer (0.2 μM), one of the species-specific primers (0.2 μM) with all the other DNA templates (10 ng of each, a total 10 ng of template) except from the corresponding one. Each reaction was performed in duplicates.

The sensitivity of the multiplex qPCR method was checked by using five 10-fold serial dilutions of each DNA template (10 ng/μL to 1 pg/μL). Each one of the serials diluted DNA templates was assayed in triplicates along with two no-template samples. The Ct of the dilutions on the dynamic ranges of each template was used to calculate the slope of the linear regression between Ct values (*y*-axis) and DNA concentrations (*x*-axis), the efficiency and the squared regression coefficient (R^2^) for each template.

## 3. Results

### 3.1. Species-Specific Primer Design and Optimization of Multiplex PCR

Since the designed primers are based on diagnostic SNPs, theoretically, the analysis could be influenced by potential intraspecific polymorphisms at the 3′ end region of the species-specific primers. Nevertheless, in the vast majority (317 out of the 318) of the specimens tested, one single band was produced, in agreement with the expected products (Table 2, Figure 4), indicating the lack of genetic diversity at the 3′ end of the primers. The single exception was a *M. barbatus* sample that was identified as *M. surmuletus* with both genes. In order to validate the technique, this sample was sequenced and indeed was confirmed as *M. surmuletus* and was therefore erroneously assigned as *M. barbatus*, probably not because of intended commercial fraud but due to its small size. Apart from this, all diagnostic bands were equally intense and clear for all species. Furthermore, all 10 fried *M. barbatus* samples analyzed were correctly identified producing the correct band in agarose gel, whereas no band was observed in the five *Trachurus* sp. specimens tested. Thus, the Multiplex PCR presented can reliably identify the four Mullidae species under investigation. It should be also noted that it is of high importance to include an internal positive control to assess the quality of the DNA template, even though we have not encountered any issues with this so far in all the analyzed fish specimens. However, adding another primer to the existing mix of five could increase the likelihood of primer-dimers and the generation of false products, which can be complex to manage. Instead, as an alternative solution, a second reaction should be prepared, with either CO1 or CYTB universal primers, resulting in the amplification of 655 bp and 464 bp products, respectively. This additional step ensures that the quality of the DNA template is appropriate for PCR. It is also recommended to perform each reaction in duplicates to minimize the possibility of technical errors.

Overall, the multiplex PCR with the five primers is completed within approximately 80 min and the total cost of the method does not exceed the cost of a simple PCR, avoiding the use of expensive and time-consuming restriction enzymes and sequencing. The results obtained from the 318 samples analyzed did not show any amplification failure. Thus, all tested samples were therefore assigned to the correct species in full agreement with the results from Sanger sequencing. Based on comparison with Sanger sequencing, both multiplex PCR are characterized by specificity and sensitivity equal to 1. Moreover, fried samples were identified as *M. barbatus* in accordance to the restaurants’ reports.

### 3.2. Optimization of Multiplex Real-Time PCR of CO1

Furthermore, we redesigned some of the primers of the CO1 gene to be compatible with Real-Time PCR (Table 2), in order to exploit the velocity and the simplicity of a multiplex qPCR method. The PCR products are designed to have both distinct molecular weight and Tm. The length of the expected PCR amplicons and the corresponding Tm values as determined via the Melt-Curve analysis are shown in Figure 5. Tm values of the species-specific products differ by at least 1 °C, which allowed us to easily identify each one of the four species via the Melt–Curve analysis. Gel electrophoresis of the species-specific PCR amplicons is presented in Figure 6. In all cases, the multiplex reactions for each species yielded an identical melting curve to the reaction in which a single set of primers was used, indicating that the primer mix is appropriate for the amplification of any of the four species in a multiplex reaction. Since the PCR products corresponding to different species have different lengths, gel electrophoresis analysis can be performed to confirm the identification of each species. The set of multiplex primers was submitted to specificity tests to eliminate possibility of cross-reactivity that could lead to false positive results. Each primer set was used in a reaction containing DNA templates derived from all the studied species except that one corresponding to the particular primer set. No PCR product was produced under 32 Ct (negative control) as all sets of primers yielded a fragment of the expected Tm value only when the specific template was present (see Melt–Curve analysis in Figure 5). Moreover, the reactions containing no template and the reactions with all the templates except from the corresponding one, yielded one peak at 72.5 ± 0.5 corresponding to dimers created by 2COIMsR with the forward primer. All sets of primers yield one product exclusively in the presence of the specific template. Gel electrophoresis of the reaction products confirmed this conclusion as the fragment lengths detected corresponded with the Melt–Curve analysis peaks in all cases (Figure 5). Furthermore, we used five 10-fold serial dilutions of each DNA template (10 ng to 1 pg) to evaluate the analytical sensitivity of single and multiplex qPCR. The multiplex qPCR linear dynamic ranges of each DNA control sample were found to be 10 ng to 1 pg for all the species, except of *M. surmuletus*, in which at least 10 pg are needed for a confident result (Figure 7). Last but not least, in agreement with the multiplex PCR, 317 out of the 318 samples investigated were assigned according to the morphological identification. As explained earlier, the single case of “misassignment” was not due to a methodological error but it could be attributed to misinterpretation due to false information from the sample collectors, probably due to the small size of the specimen, less than 10 cm in length.

## 4. Discussion

Commercial fraud of food products includes the set of practices conducted either intentionally with the aim of misleading traders, such as businesses and consumers, or unintentionally due to ignorance of the scientific names. In any case, it is a major problem for European seafood products hampering sustainable production of marine products [20]. Particularly, on account of the notably great biodiversity of both native and introduced tropical fishes in the Mediterranean, there is a huge possibility of finding hardly identifiable fish species in the markets of Mediterranean countries leading to mislabeling [21]. Herein, we describe the development and validation of a set of novel molecular techniques towards the identification of goatfishes (family Mullidae), a high valued group of fishes that are listed among the highest-preferred ones by the consumers in Mediterranean countries [5,6]. More specifically, the developed methodologies have the ability to distinguish the two native species in the Mediterranean species of the Mullidae family, i.e., *M. barbatus* and *M. surmuletus*, that are of great commercial interest, against the imported species of lower economic value *P. prayensis* as well as the Lessepsian migrator *U. moluccensis,* which is of lower value but ecological importance due to antagonism with local populations. The methods can be completed in less than 120 min, or 5 h for 50–60 samples running simultaneously, and can be also applied in tissue samples originating from cooked samples. The applicability of these cost-effective, quick, and easy-to-implement methodologies is expected to contribute to the detection and elimination of intentional or unintentional trade fraud.

The aim of the present study was to develop a fast, reliable, and low-cost method for the identification of the fish species within the family Mullidae traded in the Eastern Mediterranean. There are numerous techniques, such as microsatellite markers, LAMP, AFLP, RFLP, SSCP, multiplex PCR, Taqman PCR, and Melt–Curve qPCR, all widely used in genetic diversity analyses, identification of polymorphic loci, and molecular identification of various organisms, each of which characterized by strengths and weaknesses [1].

AFLP is a PCR-based DNA fingerprinting technique that uses selective amplification of restriction enzyme-digested DNA fragments to generate a high-resolution fingerprint of an organism’s genome. It is most suitable when there is no prior knowledge of the sequence, i.e., when other more targeted methods are not applicable. However, it requires more specialized equipment and reagents while the protocol is more time consuming, making it less attractive for the purpose of our study. Microsatellite markers are short tandem repeat (STR) markers consisting of repeated DNA sequences that are highly polymorphic. It can be relatively inexpensive compared to other molecular methods and are used mostly for genetic mapping, population studies, and forensic analysis. Both of these methods are used for the study and the detection of new genetic markers, but are not suitable for fast molecular identification. Contrary to AFLP and STR techniques, our study capitalized the knowledge of DNA sequence information of two well-known genetic markers, CO1 and CYTB that we generated via Sanger sequencing.

Furthermore, based on the characteristics of genetic markers, many methods such as LAMP, RFLP, SSCP, multiplex PCR, Taqman Assay and Melt–Curve qPCR can be used to distinguish species. LAMP is a nucleic acid amplification technique that can amplify a specific DNA sequence under isothermal conditions, so it is relatively low cost and often used in resource-limited settings for point-of-care diagnostics. However, the combination of LAMP and multiplex is complicated and is more labor intensive. RFLP uses restriction enzymes to cleave DNA at specific sites, generating fragments of varying lengths that can be separated using gel electrophoresis. Compared to multiplex PCR, it requires at least two enzymatic reactions (PCR and Digestion(s)) increasing the total duration of the procedure. Moreover, a primary analysis demonstrates that CO1 and CYTB markers are not suitable to apply RFLP for the four species, as the only available suitable restriction sites require digestion with enzymes that are uncommon and relatively costly. SSCP is a technique that separates single-stranded DNA fragments based on their conformation, which is influenced by their nucleotide sequence. Furthermore, the results can be affected by the presence of secondary structures in the DNA. The resolution and sensitivity of the technique can be affected by many technical factors, which can make it difficult to achieve consistent and reproducible results.

Techniques such as RFLP, LAMP, and SSCP can be relatively easier to design and may use inexpensive consumables compared to multiplex PCR and Melt–Curve PCR analysis; however, in terms of labor and time demands, multiplex PCR and Melt–Curve qPCR present many advantages. Multiplex PCR allows the amplification of multiple target sequences in a single reaction, which is useful for interrogating multiple species simultaneously. Although multiplex PCR and Melt–Curve PCR analysis are tedious to design, they are simple to apply and allow very reliable interpretation of the results. The Taqman Assay, on the other hand, is an expensive expansion of the real-time PCR technique that is very accurate for the quantification but it cannot be used to detect multiple species simultaneously. Thus, we chose to develop two multiplex PCR and one Melt–Curve qPCR as a diagnostic tool to identify species of the Mullidae family.

It should be noted that, despite the wide geographic range of our collections (Table 1, Figure 1), the developed techniques failed to identify specimens of any of the two *Mullus* species at a population level, on account of the lack of population-specific SNPs. This fact can be attributed to the biology of these fish species that constitute pelagic spawners [8,22]. Their early life cycle begins with extended pelagic stages as larvae and juveniles, lasting for a few weeks, a period during which they may spread influenced by sea currents, and only post metamorphosis, they remain in certain benthic substrates throughout the rest of their lives [23]. Nevertheless, according to our results, in agreement with personal communication with local fishermen from Romania and Bulgaria, *M. surmuletus* is not present in the Black Sea. Hence in this case, the identification of any *M. barbatus* samples in any Mediterranean country would confirm their origin from the Mediterranean Sea a priori. As it concerns the Lessepsian invasions from the Red Sea towards the Mediterranean, it should be highlighted that although two species belonging in the family Mullidae are supposed to have invaded (*U. moluccensis* and *U. pori*) [14,24], all collected samples were identified as *U. moluccensis*, whereas no *U. pori* was detected, questioning the population settlement status of the latter in the Greek waters, as well as the success of this invasion.

## 5. Conclusions

In conclusion, the developed methodologies described in the current study constitute a valuable set of techniques for the detection of commercial fraud in the seafood industry. They avoid the need of highly sophisticated equipment and can be ranked as cost effective, providing rapid and reliable results. Their application could be very useful for public and private laboratories in an attempt to prevent mislabeling of fish within the family Mullidae traded in Eastern Mediterranean countries. Apart from the food industry, they can provide valuable validation check points for future ecological, biological, and early life history studies of fish within this family.

## Figures and Tables

**Figure 1 genes-14-00960-f001:**
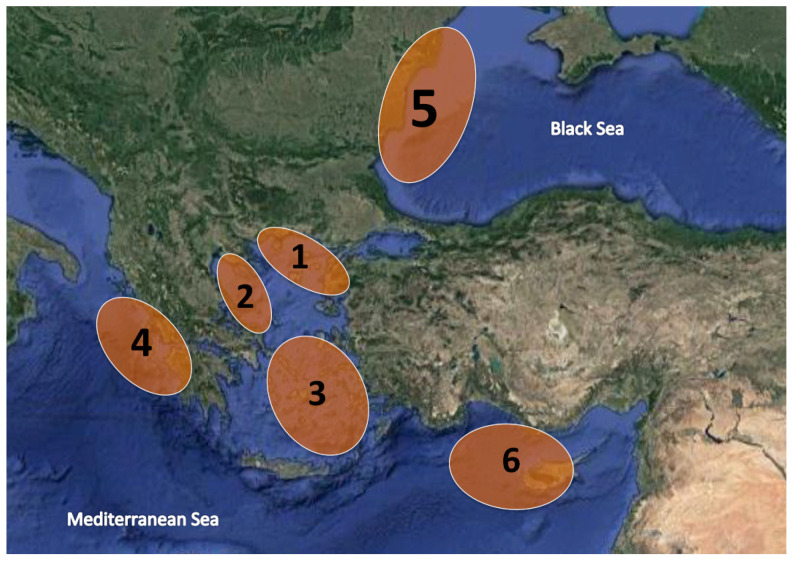
Sampling locations in the Eastern Mediterranean and Black Sea. Numbers 1–6 correspond to marine area names as mentioned in Table 1.

**Figure 2 genes-14-00960-f002:**
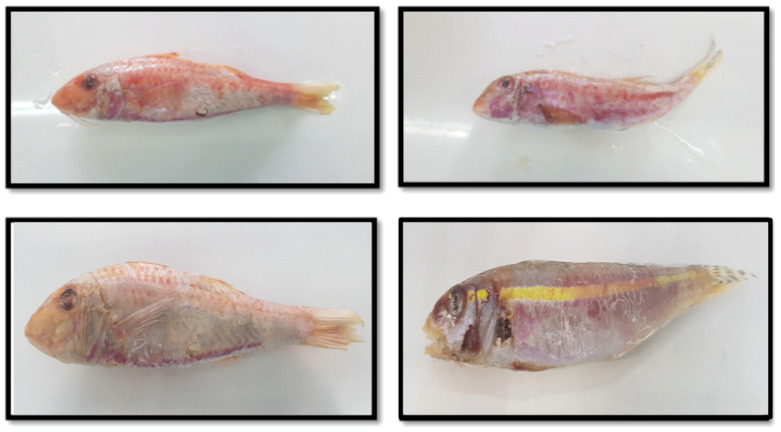
Morphological features of *Mullus barbatus* (upper left), *Mullus surmuletus* (upper right), *Pseudupeneus prayensis* (down left) and *Upeneus moluccensis* (down right).

**Figure 3 genes-14-00960-f003:**
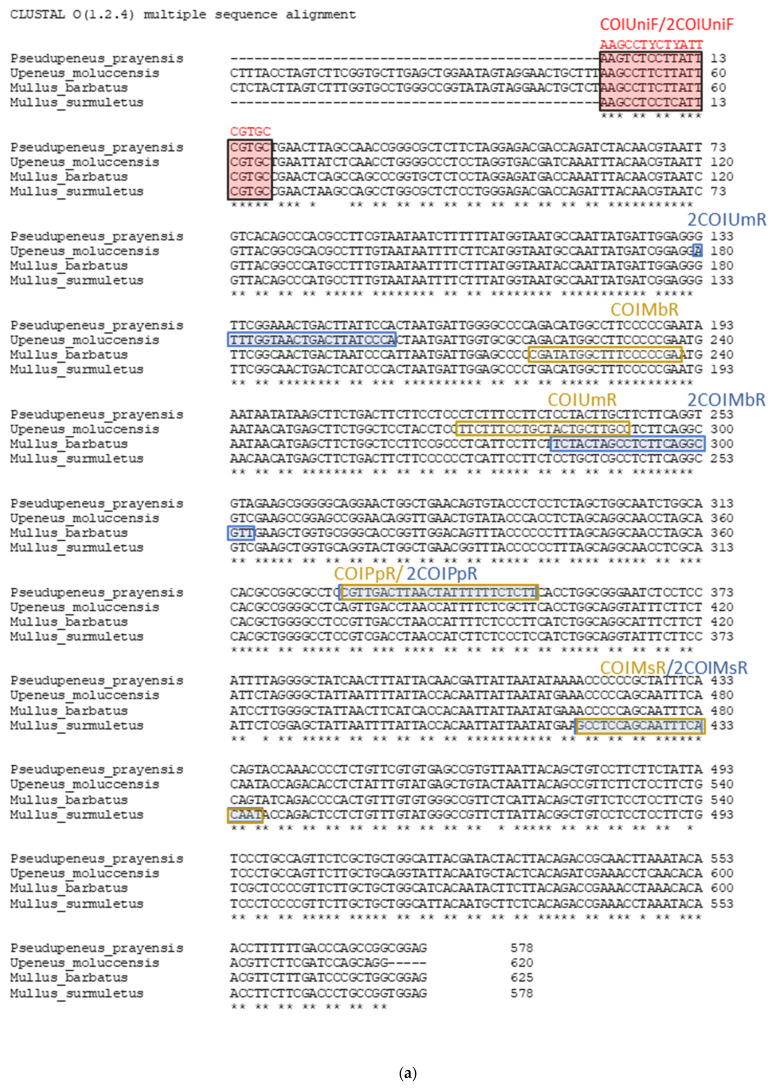
(**a**) Multiple Sequence alignment of the CO1 region of the four *Mullidae* species. *P. prayensis*; *U. moluccensis*; *M. barbatus*, and *M. surmuletus*. The red rectangle highlights the position of the forward universal primer; the yellow, the species-specific reverse primers used in multiplex PCR; and the blue, the species-specific reverse primers used in Multiplex Real-Time PCR. The exact sequence of each primer is shown in Table 2. The 3′ end of each species-specific primer was designed to target the diagnostic SNP of each species. No intra-specific variation was detected at these SNPs in the analyzed samples. (**b**) Multiple Sequence alignment of the Cyt b region of the four *Mullidae* species. *P. prayensis*; *U. moluccensis*; *M. barbatus*, *M. surmuletus*. Similar to CO1, no intra-specific variation was detected at these SNPs in the analyzed samples.

**Figure 4 genes-14-00960-f004:**
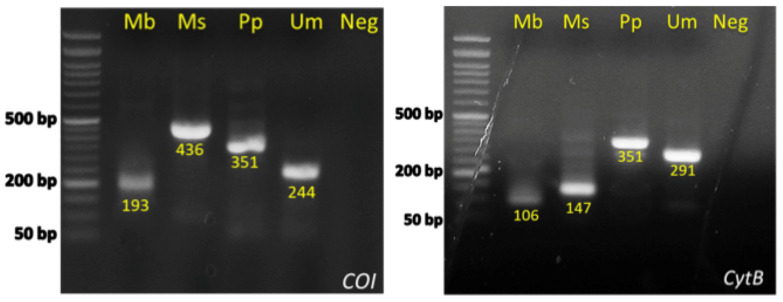
Agarose gel electrophoresis depicting the two multiplex PCR results, i.e., CO1 (**left**) and CYTB (**right**) for the species *M. barbatus* (Mb), *M. surmuletus* (Ms), *P. prayensis* (Pp), and *U. moluccensis* (Um).

**Figure 5 genes-14-00960-f005:**
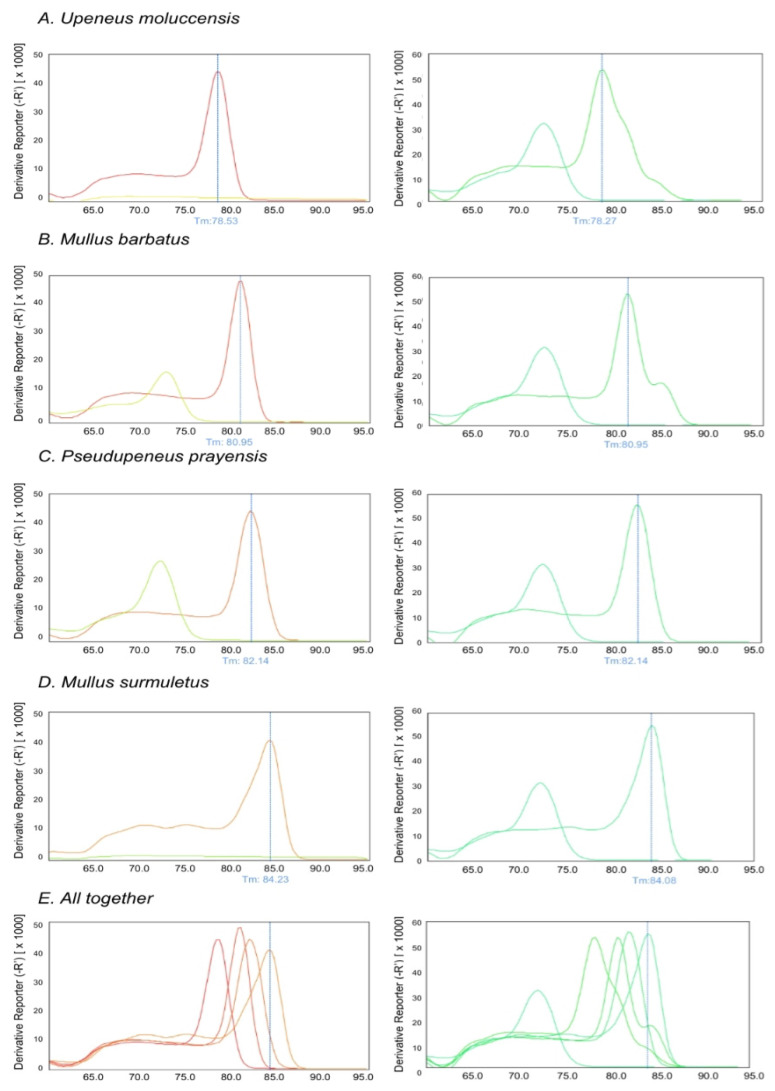
Melt–Curve analysis of Real-Time PCR reactions of specific primers with the corresponding template, with all other templates except from the corresponding one and without any template (1) and multiplex Real-Time PCR reactions (2) for species *U. moluccensis*. (**A**), *M. barbatus* (**B**), *P. prayensis* (**C**), and *M. surmuletus* (**D**). Melt curves demonstrated all at once for every species in (**E**).

**Figure 6 genes-14-00960-f006:**
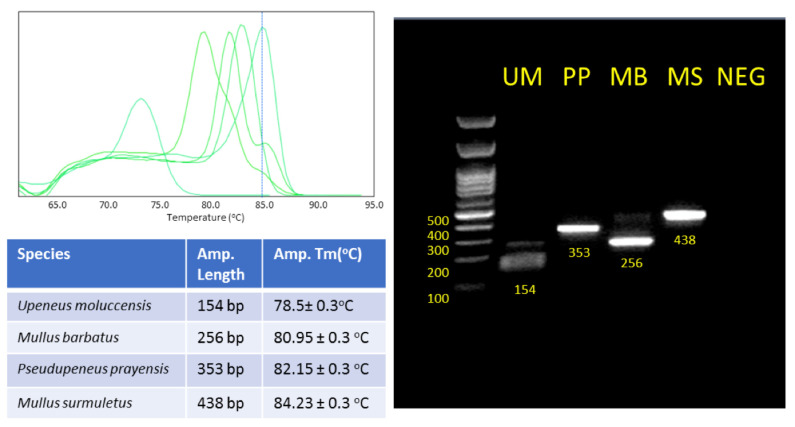
Gel electrophoresis of the species-specific PCR amplicons produced using real-time PCR.

**Figure 7 genes-14-00960-f007:**
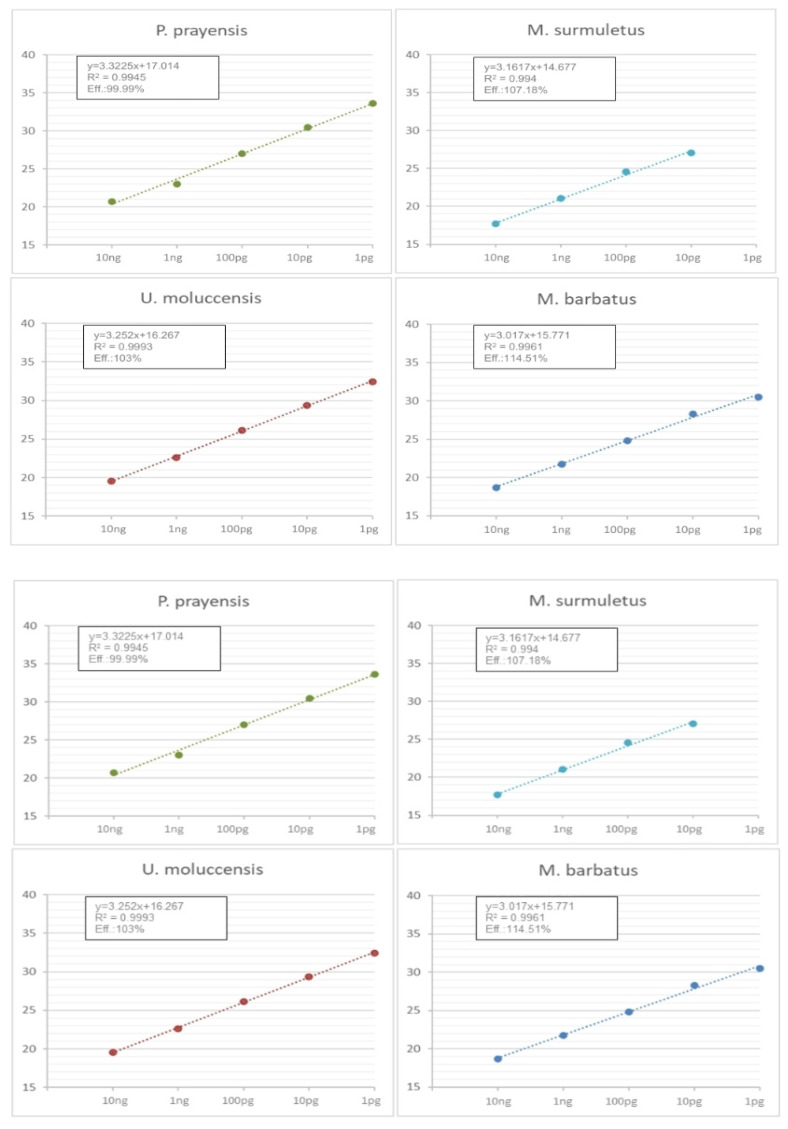
Standard curve diagram of real-time PCR reactions using serial template dilutions (10 ng to 1 pg) amplified with the specific set of primers, indicating the amplification efficiency of the different primer pairs.

**Table 1 genes-14-00960-t001:** Number of sampled specimens per species and sampling location. Numbers in the parentheses next to each marine area are in accordance to the map of Figure 1.

*Marine Area/Species*	*Mullus surmuletus*	*Mullus barbatus*	*Upeneus moluccensis*	*Pseudupeneus prayensis*
**Northeast Aegean (1)**	34	30	-	-
**Northwest Aegean (2)**	31	30	-	-
**Central Aegean (3)**	12	18	-	-
**Ionian Sea (4)**	21	5	-	-
**Black Sea (5)**	-	62	-	-
**Cyprus (6)**	20	20	15	
**Atlantic (FAO34) (7)**	-	-	-	20
**Total**	118	165	15	20

**Table 2 genes-14-00960-t002:** The species-specific designed primers.

*Name*	Sequence	Tm	Amplicon Length	Gene Target	Targeted Organism
*Primer Mix for Multiplex CO1*					
*COIUniF*	AAGCCTYCTYATTCGTGC	60.18	-	CO1	Universal
*COIMbR*	TTCGGGGGAAAGCCATATCG	62.54	193	CO1	*Mullus barbatus*
*COIUmR*	GGCAAGCAGTAGCAGGAAAGAA	63.62	244	CO1	*Upeneus moluccensis*
*COIPpR*	AAGAGAAAAAATAGTTAAGTCAACG	56.59	351	CO1	*Pseudupeneus prayensis*
*COIMsR*	ATTGTGAAATTGCTGGAGGC	59.58	436	CO1	*Mullus surmuletus*
*Primer Mix for Multiplex CYTB*					
*Cyt bUni/H15149*	AAACTGCAGCCCCTCAGAATGATATTTGTCCTCA	71.01	-	CYTB	Universal
*CyMbF*	GTAGGCGTTRTTCTTCTTCTG	59.41	106	CYTB	*Mullus barbatus*
*CyPpF*	TCGGCTCACTACTTGGGCTA	63.01	351	CYTB	*Pseudupeneus prayensis*
*CyMsF*	GCCTCTACTACGGCTCATAT	58.64	147	CYTB	*Mullus surmuletus*
*CyUmF*	TGCACTACACATCAGACATT	57.32	291	CYTB	*Upeneus moluccensis*
*Primer Mix for Melt–Curve real-time PCR CO1*					
*2COIUniF*	AAGCCTYCTYATTCGTGC	60.18	-	CO1	Universal
*2COIUmR*	TGGGATAAGTCAGTTACCAAAT	54.7	154	CO1	*Upeneus moluccensis*
*2COIMbR*	AACGCCTGAAGAGGCTAGTAGA	60.3	256	CO1	*Mullus barbatus*
*2COIPpR*	AAGAGAAAAAATAGTTAAGTCAACG	54.8	353	CO1	*Pseudupeneus prayensis*
*2COIMsR*	ATTGTGAAATTGCTGGAGGC	55.3	438	CO1	*Mullus surmuletus*

## Data Availability

The data presented in this study are openly available in GenBank NIH genetic sequence database under the accession numbers OK247993-OK248002.

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
