# Peer review of "Development of Multiplex PCR and Melt–Curve Analysis for the Molecular Identification of Four Species of the Mullidae Family, Available in the Market"

_genes, 2023, doi:10.3390/genes14050960_

Round 1

Reviewer 1 Report

Based on the importance of authentication of food products and verication of their identity, in this manuscript, authors designed two multiplex PCR assays and one multiple melt curve analysis real time PCR for identificating four species of the Mullidae family in the market. Overall, the research presents a solid piece of work, the methodology is adequate to the proposed objectives, and the conclusions are supported by the results. However, the authors also should go over the manuscript carefully, especially, the experimental design should be further clarified before publication.

1. As the authors have mentioned in the MS, if the designed primers are based on the diagnostic SNPs, theoretically the analysis could be influenced by potential intraspecific polymorphisms at the 3’end region of the species-specific primers. Accordingly, the detaild SNPs sequences as well as the designed primers should be provided in the manuscript to help the readers better understand the methodology or the experimental design of the MS.

2. The amplification efficiency of primer pairs is very important for the PCR especiall for the real-time PCR. Generally, the amplification efficiency of different target genes in the PCR detection system should be equivalent. I wonder whether the authors tested the amplification efficiency of different primer pairs. It is better to provide the amplification efficiency in the Figure 6.

3. Generally speaking, internal reference is usually necessary for the PCR detection system to verify the validity of PCR amplification system. For example, RNase P was used as the internal reference for all the experiments during the detection of COVID-19. Similarly, authors are strongly recommended to add the internal reference.

4. When conducting the traditional PCR detection followed by gel electrophoresis, it is easy to cause false positive problem caused by aerosol pollution. Actually, Figure 6 showed that the detection sensitivity was extremely high. Accordingly, there is a high probability of false positive in the detailed detection. This method is not recommended, If possible, the authors are strongly recommended to use Taqman PCR method.

5. The authors are strongly recommended to add more gDNA templates from common economic fish in the market to validate the specificity of species-specific primers.

6. As the SNP may change frequently, accordingly, the stable and long DNA sequence differences are better than SNPs in species-specific primers screening. Authors are recommended to sequence the genomic DNA and find the suitable species-specific sequences for developing the TaqMan real-time PCR method.

Author Response

Based on the importance of authentication of food products and verification of their identity, in this manuscript, authors designed two multiplex PCR assays and one multiple melt curve analysis real time PCR for identificating four species of the Mullidae family in the market. Overall, the research presents a solid piece of work, the methodology is adequate to the proposed objectives, and the conclusions are supported by the results. However, the authors also should go over the manuscript carefully, especially, the experimental design should be further clarified before publication.

Response: We are grateful to the first reviewer for recognizing the importance of our work as well as the validity of the developed methodologies. We believe and hope that after incorporating the proposed modifications and addressing all reviewers’ suggestions, the experimental design is now further clarified and the manuscript has been deeply improved.

  1. As the authors have mentioned in the MS, if the designed primers are based on the diagnostic SNPs, theoretically the analysis could be influenced by potential intraspecific polymorphisms at the 3’end region of the species-specific primers. Accordingly, the detaild SNPs sequences as well as the designed primers should be provided in the manuscript to help the readers better understand the methodology or the experimental design of the MS.

Response: In accordance to the reviewer’s comment, as well as in an effort to help the readers better understand the methodology the detailed sequence fragments of CO1 and CYTB were added in two separate figures in the revised manuscript, indicating the SNPs where the diagnostic primers were designed (please see newly modified Figures 3a and 3b). We should also emphasize that no intra-specific variation was detected at these SNPs.

  1. The amplification efficiency of primer pairs is very important for the PCR especiall for the real-time PCR. Generally, the amplification efficiency of different target genes in the PCR detection system should be equivalent. I wonder whether the authors tested the amplification efficiency of different primer pairs. It is better to provide the amplification efficiency in the Figure 6.

Response: As the reviewer mentioned, the amplification efficiency is a very important parameter, which was indeed tested for the different primers. Following her/his suggestion, Figure 6 (Figure 7 in the revised manuscript), was replaced by a new one to include amplification efficiency.

  1. Generally speaking, internal reference is usually necessary for the PCR detection system to verify the validity of PCR amplification system. For example, RNase P was used as the internal reference for all the experiments during the detection of COVID-19. Similarly, authors are strongly recommended to add the internal reference.

Response: We recognize the importance of including an internal positive control to assess the quality of the DNA template, even though we have not encountered any issues with this so far in the analyzed fish specimens. However, adding another primer to the existing mix of five could increase the likelihood of primer-dimers and the generation of false products, which can be complex to manage. Instead, as an alternative solution, a second reaction should be prepared, with either CO1 or CYTB universal primers, resulting in the amplification of 655 bp and 464 bp products, respectively. This additional step ensures that the quality of the DNA template is appropriate for PCR. As always, we recommend performing each reaction in duplicates to minimize the possibility of technical errors. Following the reviewer’s suggestion, this part was added in the “Results” of the revised manuscript (Section 3.1).

  1. When conducting the traditional PCR detection followed by gel electrophoresis, it is easy to cause false positive problem caused by aerosol pollution. Actually, Figure 6 showed that the detection sensitivity was extremely high. Accordingly, there is a high probability of false positive in the detailed detection. This method is not recommended, If possible, the authors are strongly recommended to use Taqman PCR method.

Response: Indeed, aerosol contamination e.g. during DNA extraction or the PCR reaction preparation can affect the results and this would be impossible to avoid in end-point PCR reactions. However, our system is very robust for the following reasons: proper positive and negative controls are included, reactions are carried out in technical replicates for each sample, and, most importantly  we use a relatively high amount of template (typically 10-20 ng of DNA template is utilized per reaction) well above the detection threshold, so that the Ct of the reaction should be as low as 20, similar to the positive controls. In case of aerosol contamination the Ct in the PCR would be much higher, so there is a built-in way to discriminate false positive results due to aerosol contamination. Finally, it should be emphasized that our goal was to create a cost-effective method for Mullidae identification. Taqman MGB probes are by far more expensive and Taqman reactions can also be affected by cross-contamination. Only if a very large number of samples are regularly examined, Taqman PCR could be characterized as cost-effective. Therefore, taking into consideration that the number of samples intended to be tested is relatively low, we chose to use traditional multiplex PCR and multiplex qPCR combined with Melt Curve Analysis.

  1. The authors are strongly recommended to add more gDNA templates from common economic fish in the market to validate the specificity of species-specific primers.

Response: According to the reviewer’s suggestion, gDNA from five Trachurus sp. was included in the analyses to validate the specificity and the results verified the specificity of the designed methodologies (Please see sections 2.1 and 3.1).

  1. As the SNP may change frequently, accordingly, the stable and long DNA sequence differences are better than SNPs in species-specific primers screening. Authors are recommended to sequence the genomic DNA and find the suitable species-specific sequences for developing the TaqMan real-time PCR method.

Response: The design of our primers is based on sequences obtained from 318 specimens, from which we have identified species-specific polymorphisms that are 100% stable across each species. As shown in Table 1 and Figure 1, specimens of each species were collected from different geographical regions, allowing us to detect intra-species SNPs. Our primers have been designed to target regions containing species-specific polymorphisms while avoiding intra-specific polymorphisms. It should be noted that the same regions would be used for the development of a TaqMan assay thus again may change frequently, however in this case our assay would be much more expensive, which falls out of the purpose of our study.

Author Response

  1. The numerical details of the results should be included in the abstract.

Response: Following the reviewer’s comment, further details concerning the results were added in the abstract of the revised manuscript

  1. The scientific names of fish in keywords should be in italics.

Response: Corrected according to the reviewer’s comment

  1. Some words used, such as mislabel vs mis-label, were not consistent

Response: Corrected in the whole manuscript, according to the reviewer’s comment

  1. Line 95, Pseudupeneus prayensis, please check the spelling and the scientific name should be in italics.

Response: Corrected according to the reviewer’s comment

  1. Line 108, the first sentence should be revised.

Response: The sentence was modified, as suggested by the reviewer, as follows: “Thus, user-friendly molecular techniques that could be implemented in standard not sophisticated equipment would be very helpful towards fast routine identification of large numbers of food product particles”

  1. I think the image of Pseudupeneus prayensis in figure 2 was retrieved from the internet source (https://www.newsea.dk/products/item/pseudupeneus-prayensis-mauritania). Therefore, the reference should be added.

Response: As correctly mentioned by the reviewer, Pseudupeneus prayensis was retrieved from the internet. Therefore, the Figure 2 was accordingly modified, providing an image from our collections.

  1. Cyt b vs cyt b?, CO1 vs COI?

Response: In accordance to the reviewer’s comment, all the gene names were written with capitals with the same way in the revised manuscript.

  1. Line 148, DNA extraction was repeatedly mentioned in the topic.

Response: The title (2.1) was corrected following the reviewer’s comment

  1. The list of primers in table 2 should be demonstrated as pairs of primers. Moreover, the intentionally identified species should be indicated for each primer pair.

Response: The intentionally identified species is now indicated for each primer pair in the revised manuscript, as suggested by the reviewer (please see the revised Table 2). However we cannot present the primers as pairs because they are utilized in a mix, as described in section 2.2

  1. Lines 198-200, oC vs oC?

Response: Corrected according to the reviewer’s comment

  1. The results of the specificity test were not shown.

Response: Specificity (true negative rate) is the probability of a negative test result, conditioned on the individual truly being negative. As no negative test was revealed, specificity test was equal to 1. These results were added in the section 3.1 of the revised manuscript

  1. The sensitivity test should be conducted

Response: Analytical sensitivity or limit of detection refers to the minimum number of nucleic acid copies in a sample that can be detected. Furthermore, sensitivity (true positive rate) is the probability of a positive test result, conditioned on the individual truly being positive. Sensitivity of the multiplex PCR was equal to 1, whereas amplification efficiency was also run and results are shown in the revised manuscript (please see section 3.1 and Figure 7)

  1. Where are the results from the cooked sample?

Response: As correctly mentioned by the reviewer, results are now provided in the revised manuscript (please see section 3.1)

  1. Mix samples should be also used for multiplex PCR analysis.

Response: Mix samples would be more meaningful if the species of interest are found in mashed form in the market. Most of the times, our specimens have origin from one whole fish or a part of it, such as head, fin, tail, bones etc. Thus, the origin of the DNA is from a single fish, and the expected result would be one band in the gel or one curve in the qPCR. In case of cross-contamination during DNA extraction or if the PCR preparation is part of good laboratory practice and not the assessment of PCR method. Also, if there is contamination, then in case of multiplex PCR, more than one band would be shown in the gel, and in case of qPCR the curve would be wider and unclear and the test would provide no accurate results.

  1. The results were not appropriately analyzed and demonstrated

Response: We hope that after addressing all the comments and suggestions provided by the reviewers, the results are now presented more appropriately and this section has been substantially improved.

  1. Grammatical errors, inconsistency of the word used, and mis-spelling were found throughout the manuscript.

Response: The whole manuscript has been deeply revised and inconsistencies as well as mis-spellings have been corrected.

Reviewer 3 Report

MS entitled “Development of multiplex PCR and melt curve analysis for the molecular identification of four species of the Mullidae family available in the market” is not acceptable in the present format. I criticize the authors at the outset for poorly written Discussion portion. I ask the authors multiplex PCR and one real time PCR melt curve analysis has become the simplest tool? Please justify your statement after going through the article of Kotasanopoulos et al., 2020. I feel that, there are so many molecular tools and proteomics based analysis by which identification of fish species is possible. To mention a few are: LAMP, microsatellite markers, AFLP, RFLP, SSCP. Please mention the tests in Discussion and compare with your developed tests.

My second observation: Introduction portion of the MS is not up to the mark. I expect the authors will add one paragraph on Mullidae family and details of geographical distribution of the genera. I also advise the authors to delete tables 1 & 2 and please mention the same in supplementary file.

Therefore, I feel the MS needs major revision with more scientific justifications.

Author Response

MS entitled “Development of multiplex PCR and melt curve analysis for the molecular identification of four species of the Mullidae family available in the market” is not acceptable in the present format. I criticize the authors at the outset for poorly written Discussion portion.

Response: We would like to thank the reviewer for evaluating our manuscript and for the useful comments. We believe and hope that after implementing and addressing all three reviewers’ comments and suggestions the quality of the written parts has been improved and now meets the standards of the journal Genes.

I ask the authors multiplex PCR and one real time PCR melt curve analysis has become the simplest tool? Please justify your statement after going through the article of Kotasanopoulos et al., 2020. I feel that, there are so many molecular tools and proteomics based analysis by which identification of fish species is possible. To mention a few are: LAMP, microsatellite markers, AFLP, RFLP, SSCP. Please mention the tests in Discussion and compare with your developed tests.

Response: In response to the reviewer’s question, indeed in our experience multiplex PCR and real time PCR melt curve analysis proved the easiest-to-perform and simplest molecular diagnostic tools in terms of cost efficiency and time demands. Further, we tried to search the scientific literature and we were not able to locate the paper Kotasanopoulos et al., 2020. We suppose that the reviewer probably meant Kotsanopoulos et al. 2021. Taken into consideration this assumption, in line with the reviewer’s suggestion, we went through the article authored by Kotsanopoulos et al. 2021, as well as similar articles, and added a detailed paragraph in the Discussion regarding the comparison of different molecular techniques and their use in species identification. However, we do feel that the focus of our analysis is diluted. In any case, it is impossible to achieve an in-depth comparison of all available techniques. Actually such a task could be a very nice topic for a thorough review article. The technique of choice depends on the available infrastructure and lab equipment, the cost and time limits, as well as the reliability (specificity and sensitivity) required. We have explained clearly in the manuscript that our intensions were to develop a methodology that combines reliability with low-cost/time demands. In this vein, we believe that the presented methodology meets very successfully our set aims. Although we have included four paragraphs in the Discussion Section, we would like to politely urge the Editor to allow us the option to withdraw them if the Referee #3 agrees.

My second observation: Introduction portion of the MS is not up to the mark. I expect the authors will add one paragraph on Mullidae family and details of geographical distribution of the genera.

Response: In accordance to the reviewer’s comment, an extra paragraph was added in the Introduction, presenting data concerning the Mullidae family and the geographical distribution of the genera within the family.

I also advise the authors to delete tables 1 & 2 and please mention the same in supplementary file.

Response: We appreciate the suggestion of the reviewer, however, in line also with the two other reviewers’ comments, we believe that Tables 1 and 2, as well the novel Figure 3, are important parts of the developed methodologies and should therefore compose part of the main text in the manuscript. Nevertheless, if the Editors propose to present these data as supplementary material, we can transfer them in supplementary files.

Therefore, I feel the MS needs major revision with more scientific justifications.

Response: We trust that after implementing the recommendations of the reviewers, our revised manuscript is improved and meets the reviewer’s expectations.

Round 2

Reviewer 2 Report

All query has been addressed. The manuscript is suitable for publication.

Reviewer 3 Report

The manuscript has been improved significantly. The concerns raised have been addressed and modifications suggested have been included in the revised manuscript. I am satisfied with the justifications given by the authors. The manuscript in revised form looks good and of interest for the readers. The manuscript may be accepted for publication.